# Physical Fitness, Screen Time and Sleep Habits According to Obesity Levels in Schoolchildren: Findings from the Health Survey of the Extreme South of Chile

**DOI:** 10.3390/ijerph192013690

**Published:** 2022-10-21

**Authors:** Fernanda Carrasco-Marín, Fanny Petermann-Rocha, Miquel Martorell, Yenny Concha-Cisternas, Solange Parra-Soto, Rafael Zapata-Lamana, Javier Albornoz-Guerrero, Guillermo García-Pérez-de-Sevilla, Maria Antonia Parra-Rizo, Igor Cigarroa

**Affiliations:** 1Centro de Vida Saludable, Universidad de Concepción, Concepción 407386, Chile; 2Centro de Investigación Biomédica, Facultad de Medicina, Universidad Diego Portales, Santiago 8370068, Chile; 3Departamento de Nutrición y Dietética, Facultad de Farmacia, Universidad de Concepción, Concepción 407386, Chile; 4Escuela de Kinesiología, Facultad de Salud, Universidad Santo Tomás, Talca 3460000, Chile; 5Pedagogía en Educación Física, Facultad de Educación, Universidad Autónoma de Chile, Talca 3460000, Chile; 6Department of Nutrition and Public Health, Universidad del Bío-Bío, Chillan 3780000, Chile; 7School Cardiovascular and Metabolic Health, University of Glasgow, Glasgow G12 8TA, UK; 8Escuela de Educación, Universidad de Concepción, Los Ángeles 4440000, Chile; 9Departamento de Educación y Humanidades, Universidad de Magallanes, Punta Arenas 6200000, Chile; 10Faculty of Sports Sciences, Department of Physiotherapy, Universidad Europea de Madrid, 28670 Madrid, Spain; 11Faculty of Health Sciences, Valencian International University (VIU), 46002 Valencia, Spain; 12Department of Health Psychology, Faculty of Social and Health Sciences, Campus of Elche, Miguel Hernandez University (UMH), 03202 Elche, Spain; 13Escuela de Kinesiología, Facultad de Salud, Universidad Santo Tomás, Los Ángeles 4440000, Chile

**Keywords:** Body Mass Index, body composition, healthy weight-obese, physical fitness, sleeping habits, screen time

## Abstract

Obesity is a worry because it is associated with a greater burden of disease, and it has been shown to be related to the health habits and physical condition of children and adolescents. Objective: To associate physical fitness, screen time, and sleep habits with the different categories of obesity in schoolchildren from the extreme south of Chile. Methods: 583 schoolchildren were included in this cross-sectional analysis. The screen time and sleep habits were measured with the Sleep Self-Report questionnaire, while the physical fitness was assessed with the Alpha Fitness test battery. The Body Mass Index/age (BMI/age) and the Waist-to-Height ratio (WtHr) were used to define adiposity using the following categories: healthy weight/low-risk waist-to-height ratio (H/LR), healthy weight/high-risk waist-to-height ratio (H/HR), overweight/low-risk waist to height ratio (O/LR), and overweight/high-risk waist to height ratio (O/HR). Results: A considerable number of schoolchildren (23.2%) presented sleep problems, while the mean screen time was 9.3 (95% CI: 8.4; 10.1) hours/day. Schoolchildren classified as H/HR showed better physical fitness than the O/HR group but worse physical fitness than the H/LR group. Conclusions: Significant differences were evidenced in the physical fitness between the adiposity categories, which could open future lines of research concerning the characterization of the healthy weight-obese adiposity categories in children.

## 1. Introduction

Childhood obesity and overweight are among the most critical public health problems worldwide. It is estimated that more than 330 million children aged 5–19 years suffer from excess malnutrition [1]. Along with this, the development of chronic non-communicable diseases is increasing at younger ages [2]. In Chile, according to the National Health Survey 2016–2017 (NHS), 39.8% and 31.2% of the general population were overweight and obese, respectively. In adolescents aged 15–19, 27.6% were overweight, and 12.2% were obese. Specifically, Punta Arenas commune (53°9′49.799″ S 70°55′1.445″ W (GPS-coordinates), where this study was carried out, belongs to an area in the extreme south of Chile, in South America continent characterized by extremely cold weather (average of 6.5 °C, with the lowest temperature of −16.4 °C in winter) [3], which has 53.8% of overweight/obese children and adolescents [4]. These alarming figures are associated, in part, with modifiable risk factors such as unhealthy eating habits, a lack of physical activity [5], increased screen time [6], and poor sleep quality [7,8].

The body mass index (BMI) is a measure of weight relative to height (weight in kilograms divided by the square of height in meters) [9]. BMI has been widely used to measure nutritional status due to its simplicity and validation in multiple studies. However, the method presents low sensitivity in evaluating adiposity [10]. In this context, several studies have observed healthy-weight individuals with diseases associated with obesity or biochemical abnormalities, while obese individuals show normal metabolic parameters, explained by the distribution of adiposity not detected through the BMI [11,12]. The BMI loses sensitivity by not discriminating between fat, muscle, and bone tissue differences or their distribution [13]. Additionally, the waist-to-height ratio (WtHr) has shown to be a highly applicable, accurate anthropometric index to measure central adiposity and a strong predictor of chronic non-communicable diseases [14,15,16]. Likewise, high physical inactivity, screen time, and low sleeping hours have been associated with childhood overweight and obesity assessed by WtHr [17]. Unfortunately, the SARS-CoV-2 lockdown exacerbated these numbers [18,19]. In 2006, the association between healthy-weight individuals with a high body fat mass was described as a sub-classification of obesity, with the term healthy-weight obese (HWO) [20]. This definition includes people with a healthy BMI but with excess fat mass. However, there is no consensus on defining the cut-off point of excessive fat mass, varying between 20–25% fat mass in men and 30–37% fat mass in women [12].

Studies have associated HWO individuals with physical inactivity, smoking, and a low-fiber diet [21,22], especially in countries with extreme cold climates, promoting physical inactivity [23]. In adolescents, previous studies have observed higher sedentary behaviors and lower physical fitness (lower endurance, lower extremity strength, and agility) in HWO compared to healthy-weight non-obese (HWNO) children [24,25,26]. Regarding sleep indicators, meeting the recommended number of hours was associated with a metabolically healthy profile, as were shorter sleep duration with higher levels of insulin, low-density lipoprotein, and C-reactive protein [27,28].

Even though the association between BMI and fat mass distribution has already been used to subclassify obesity, few studies describe their physical and lifestyle characteristics [21,22]. Few studies have focused on whether there are differences in screen time, sleep habits and physical condition in children and adolescents when classifying them according to BMI and fat mass distribution [24,25].

On the other hand, a recent systematic review indicates that overweight and obesity are more prevalent in countries with extreme cold weather [23]. In this type of climate, both physical inactivity and the development of physiological adaptations that lead to the development of obesity are favored [29]. However, there is little evidence about the association of lifestyle with BMI in schoolchildren from the south of Chile [17].

Considering these gaps, this study aimed to associate physical fitness, screen time, and sleep habits with the different categories of obesity in schoolchildren from the extreme south of Chile.

## 2. Materials and Methods

### 2.1. Design

This cross-sectional study was conducted following the STROBE guidelines [30].

### 2.2. Participants

Male and female students from the second cycle of primary education—aged 9–15 years from 3 public educational establishments—from Punta Arenas commune (Region of Magallanes and the Chilean Antarctic)—were included to participate. The sample size was calculated considering the heterogeneity of 50%, a margin of alfa error of 5%, and a confidence level of 95%. A probabilistic and stratified sample with community representativeness of 615 schoolchildren who completed all evaluations after signing the consent of the parents/legal guardian was included. A total of 14 schoolchildren were excluded for not signing the informed consent, four did not participate due to inability to perform the physical tests, and 3 for not completing all the evaluations, so the final sample was 583 schoolchildren.

The project was approved by the Ethics Committee of the south-central macro zone of the Santo Tomás University, Chile (96-20), and all procedures were carried out following the Declaration of Helsinki and Singapore.

### 2.3. Outcomes

#### Physical Fitness

*Cardio-respiratory fitness:* This was assessed using the 20-m shuttle run test (20-m SRT), also known as course-navette. It consisted of running back and forth on a 20 m track marked between two separate lines for as long as possible. The rhythm was set using audio signals. The initial speed was 8.5 km/h and was increased by 0.5 km/h intervals every 1 min. Subjects had to step behind the 20 m line when the audio signal or beep was heard. The test finished when the subject stopped because of fatigue or failed to reach the end line concurrent with the beep on two consecutive occasions. Test performance was recorded using the final speed reached (FSR) in the last stage. This test was performed using the protocol given by the Alpha Fitness test battery [31].

*Musculoskeletal fitness:* The standing broad jump (SBJ) test was used as an indicator of lower limb strength. It consisted in jumping the longest distance possible from a standing start, with both feet and swinging both arms. The distance was measured from the takeoff line to the point where the back of the heel nearest to the takeoff line landed on the ground. This test was performed using the protocol given by the Alpha Fitness test battery [31].

The handgrip strength of the dominant hand was used as an indicator of upper limb strength. (JAMAR (brand) hydraulic dynamometer (Hydraulic Hand Dynamometer^®^ Model PC-5030 J1, Fred Sammons, Inc., Burr Ridge, IL, USA). Each subject was seated in a standard position in a chair with a straight back. Students were asked to exert pressure on the dynamometer twice with each hand. To control for the effects of fatigue, the attempts were performed by alternating the hands with approximately 2 min of rest between each attempt for each hand. The best measurement was recorded for each of the two attempts [32,33].

All these tests have been widely used to assess physical fitness in children and adolescents in international and national studies [34,35].

*Self-perceived physical fitness**:* this outcome was measured with International Fitness Scale (IFIS), which was initially validated in nine European countries and languages (HELENA study) [34]. The IFIS is comprised of 5 Likert-scale questions about self-reported fitness (very poor, poor, average, good, and very good) relating to perceived overall fitness and its main components: cardio-respiratory, muscular strength, speed and agility, and flexibility. The IFIS had high validity and moderate-to-good reliability in a study with collegiate students [36]. All the information about the IFIS can be found at no cost on the website of the PROFITH research group, the original developer of this tool [37].

*Screen time:* Self-reported screen time was assessed through 3 questions, which have been used in different national and international studies: “How many hours a day do you usually watch television?” “How many hours a day do you usually play video games on a tablet, computer, or cell phone?” and “How many hours a day do you usually use a tablet, computer, or cell phone for purposes other than gaming, for example, email, chats, social networks, surf the internet or do homework?”. These questions have been used in various studies to assess the screen time of schoolchildren [26,38,39]. The average screen time was calculated by adding these three questions. Screen time was categorized as low-medium < 2 h/day and high ≥ 2 h/day, following the American Academy of Pediatrics recommendations [40].

*Sleeping habits:* The questionnaire used to assess sleep habits and problems was the Sleep Self Report (SSR) in its Spanish version. Each item has a 3-point scale to indicate the frequency of each habit: usually (2 = 5–7 times a week), sometimes (1 = 2–4 times a week), and rarely (0 = never or once a week). The questionnaire is composed of 19 items (3 of them provide additional information but are not included in any subscale), grouped into 4 subscales: (a) sleep quality, (b) sleep-related anxiety, (c) refusal to sleep, and (d) bedtime routines. An overall score is obtained by adding the scores of the 16 items (total score variable). Higher scores indicate more sleep-related problems. According to the criteria established by Orgilés and collaborators [41], the following cut-off points to indicate sleep problems were taken into account: 7 for sleep quality, 6 for sleep-related anxiety, 4 for rejection of sleep, 3 for routines to go to sleep and 16 for the total score [41]. This questionnaire has been widely used in research that analyzes the sleeping habits of schoolchildren [26,38,39].

### 2.4. Grouping Variable

*Body Max Index (BMI):* it was used to measure the nutritional status of schoolchildren by sex, considering malnutrition as a standard deviation (SD) ≥ −2, risk of malnutrition SD ≥ −1, and normality between 0.99 and −0.99 SD. For overweight and obesity, SD ≥ 1 and ≥ 2, respectively, were considered [42].

*Waist-to-height ratio (WHtR)* is a specific index based on waist circumference and height and is highly correlated with intra-abdominal visceral adiposity and cardiometabolic risk [43,44,45]. WtHr was calculated by dividing waist circumference (cm) by height (cm). Pediatric reference values are <0.47 normal adiposity; 0.47–0.50 moderate adiposity; and >0.50 excess adiposity. These reference values have already been used in previous studies [17,26].

Waist circumference was measured using the point equidistant between the last non-floating rib and the iliac crest during shallow apnea, with the children standing erect with the abdomen relaxed. Waist circumference and height were measured in centimeters and weight in kilograms following the recommendation of the National Health and Nutrition Examination Survey (NHANES) [46], and using an inelastic tape for waist circumference; a scale Seca 813 (Seca, Hamburg, Germany), for weight and stadiometer Seca 213 (Seca, Hamburg, Germany) for height.

*Obesity sub-classification:* A subclassification was created according to BMI/age and their WtHr for this study. In this way, it will be understood that a student has a healthy BMI when they have a normal weight, low-risk WtHR when they have normal and moderate adiposity and high-risk WtHr when they have excess adiposity. Thus, the schoolchildren (girls and boys) were classified using the aforementioned variables, and the following categories were created: (1) Healthy BMI/ low-risk WtHr (H/LR); (2) Overweight/low-risk WtHr (O/LR); (3) Healthy BMI/ high-risk WtHr (H/HR); and (4) Overweight/high-risk WtHr (O/HR).

In addition, sociodemographic values such as sex (male/female), age, school grade (fifth grade/six grade/seventh grade/eight grade), place of residence (urban/rural), educative center (E.C.1/E.C.2/E.C.3) and if they belong to the school integration program (yes/no) were measured.

### 2.5. Statistical Analyses

Data were analyzed with SPSS 25.0 statistical software (SPSS Inc., Chicago, IL, USA). The descriptive data were presented as mean and 95% confidence interval (95% CI) for continuous variables and frequency and percentage for categorical variables. To analyze the differences between the categorical variables, the chi-square test was used. An ANCOVA test was used to determine the mean differences between adiposity categories. The following covariables were used: gender (male/female), age, place of residence (urban/rural), educational center (EC1/EC2/EC3), grade (5th grade/6th grade/7th grade/8th grade) and school integration program (yes/no). To establish statistically significant differences between the groups, the Bonferroni post hoc test was used. The level of significance was set as *p* < 0.05.

## 3. Results

After removing participants with missing data, 583 schoolchildren (50.8% male, 12.1 (95% CI: 11.9;12.2) year were finally included. According to their obesity sub-classification, 3.6% of schoolchildren were a healthy weight but had excess adiposity, 20.8% were overweight or obese but had normal or moderate adiposity, and 51.4% were overweight or obese and had excess adiposity. Table 1 shows the socio-educational characteristics of the schoolchildren according to their obesity sub-classification.

Table 2 shows the self-perceived physical fitness measured with IFIS according to the obesity sub-classification. They perceived their general physical condition (45%), cardiorespiratory capacity (41.3%), muscular strength (45%), and speed and agility (42.9%) to be good or very good. However, a high percentage of schoolchildren (36.4%) perceive themselves as having poor or very poor flexibility. Significant differences between groups were found in general physical condition (*p* < 0.0001), cardiorespiratory capacity (*p* < 0.0001), speed and agility (*p* < 0.0001) and flexibility (*p* < 0.0001). A higher percentage of schoolchildren in the O/HR group was found to have a poor or very poor perception of their physical condition compared to the other three groups (Table 2).

Table 3 shows the schoolchildren sleeping habits and screen time according to sub-classification of obesity. Concerning the routines to go to sleep, most of the schoolchildren presented no problems with bedtime routine (86.4%), anxiety related to sleeping (86.3%), rejection related to sleeping (84.6%), and SSR total score (83.4%). However, 23% of schoolchildren indicated having problems with the quality of sleep. Moreover, screen time was ≥2 h/day in 94.4% of the schoolchildren, with an average of 9 h a day. No differences were found between the groups in sleep habits and screen time.

Table 4 shows the screen time and sleep habits according to the schoolchildren obesity sub-classification. When the ANCOVA test was performed, no significant differences were found between the groups in daily time that schoolchildren spend in front of the screen and sleep habits measured with the sleep self-report.

On the other hand, when cardiorespiratory fitness was analyzed, significant differences between groups in the standing long jump test (ANCOVA *F*(3;579) = 21.389; *p* < 0.0001) and the 20-m shuttle run test (ANCOVA *F*(3;579) = 26.747; *p* < 0.0001) were detected. Then, a deeper analysis was done with the Bonferroni post hoc test, and it was observed that the O/HR group had a lower standing long jump test performance compared with H/LR, O/LR and H/HR groups (111.2 (108.2; 114.1 95% CI) vs. 130.9 (126.0; 135.8 95% CI), 126.6 (121.8; 131.4 95% CI) and 126.6 (117.5; 135.7 95% CI), respectively. Additionally, the O/HR group showed a lower 20-m shuttle run test performance compared to the H/LR and the O/LR groups (2.50 (2.49; 2.51) vs. 2.60 (2.58; 2.62) and 2.58 (2.56; 2.60), respectively (Figure 1).

## 4. Discussion

### 4.1. Main Results of This Study

The main results of this study indicate that most schoolchildren from Punta Arenas perceive themselves to be in an acceptable or good physical condition and without problems with sleeping habits, except in the sleep quality category. However, a large percentage spends many hours per day in front of the screen. When compared to schoolchildren from Punta Arenas according to the subclassification of obesity, who were overweight or obese and excess adiposity presented a worse perception of their physical condition compared to schoolchildren who had a healthy weight but had excess adiposity or who were overweight or obese but with normal or moderate fat distribution. Concordantly, when the physical condition was analyzed in field tests, a lower performance was observed, in terms of musculoskeletal and cardio-respiratory fitness, among the O/HR group compared to the other groups.

### 4.2. How Can These Results Be Interpreted from the Perspective of Previous Studies?

Regarding screen time, it is striking that 94.4% of schoolchildren exceed the recommended daily screen time, with an average of 9 h daily. These results are consistent with studies reporting an association between high screen time and cardiovascular risk [47], unhealthy habits such as skipping breakfast and eating fast food [48] and poor sleep quality [49] in children and adolescents. However, in this study, no significant differences were found in screen time according to the subclassification of obesity. Our results are contrary to expectations, as previous studies have shown that screen time could predict the metabolically unhealthy obese phenotype in early adolescence [50]. On the other hand, our findings are in line with a recent study carried out on Chilean schoolchildren, which warned about a large number of hours a day spent in front of a screen, which was associated with poor school achievement [38].

According to our study, 23% of schoolchildren presented problems with sleep quality but without significant differences between groups. These findings have been described in other studies, where it has been shown that sleep duration and quality are more associated with weight gain independent of nutritional status. Sleep irregularity impacts hormonal and metabolic pathways, which in turn affects energy balance, resulting in body weight gain [51]. In addition, these results are consistent with a recent study carried out on Chilean schoolchildren that reported a high percentage of students with sleep problems associated with poor academic performance [52].

The findings found give clues that suggest that a high percentage of schoolchildren analyzed have unhealthy screen time usage and sleeping habits. A possible reason could be found in the climatic and geographical conditions that these schoolchildren live in. In territories with extremely low temperatures, a large part of the year favors physical inactivity, promotes sedentary behaviors such as watching television and playing video games and modifies hormonal production, such as increased ghrelin and cortisol, which increase appetite and lipid storage mechanisms [53]. Additionally, It has been reported that in cities with cold climates, daylight varies markedly along the seasons, with only 35–60 days a year of sunlight and the rest of the year with darkness for much of the day, with a negative impact on the quality of sleep and probably in mental health [29].

Our most relevant findings concerning physical fitness were that O/HR schoolchildren had a lower performance in the standing long jump test compared to the three groups and a lower cardiovascular capacity compared to the H/LR and O/LR schoolchildren. These results are consistent with previous studies that had shown children’s tendency to be less active when they had greater adiposity [54]. However, another study showed that obese schoolchildren had lower performance in aerobic capacity, agility, lower limb muscle power, and balance compared to healthy-weight schoolchildren [55]. Therefore, it is crucial to generate a correct sub-classification in terms of obesity since there seems to be no doubt about the association of adiposity in schoolchildren with cardiorespiratory and muscular fitness compared to their non-obese healthy-weight peers [25].

### 4.3. Strengths and Limitations

The present study is a pioneer in Chile and South America in characterizing the lifestyle and physical fitness of schoolchildren from territories with extremely cold climates. In addition, according to the literature studied, it is the first study to compare self-perceived physical fitness, cardiorespiratory and musculoskeletal fitness, screen time, and sleep habits between different obesity classifications in schoolchildren from a city with extremely cold climates.

However, this study has limitations. We used the waist-to-height ratio (WtHr) and BMI to measure adiposity. Although waist circumference and BMI are used to measure visceral adiposity [15] and nutritional status, respectively [9], future studies could use more precise assessments to determine adiposities, such as dual-energy x-ray absorptiometry (DXA) or bioelectrical impedance analysis (BIA). Besides, the variables of screen time and perception of physical fitness used self-reporting instruments so that the responses of the schoolchildren were subject to their mood, understanding, and even the maturity to understand the relevance of their responses. Nevertheless, the teachers who used the surveys answered all the children’s doubts. This method of collecting information is widely used in the literature [17] and allows for collecting data from a large sample or data of a city or region in a short time. In addition, the study design only allows comparing physical fitness, screen time, and sleep habits between different obesity classifications but does not imply causality. Finally, although an analysis of covariance (ANCOVA) was performed using socio-educational variables as covariates, there may be other variables that were not measured, which could modulate the results (for example, type of diet, physical activity levels, cell phone use, among others).

### 4.4. Practical Implications and Future Lines of Investigation

This study provides national information on screen time, sleep habits, and physical condition in schoolchildren living in a city with extremely cold climates, becoming the most up-to-date evidence that accounts for sleep quality and high screen time in schoolchildren from Chilean public educational centers. This study also exposes the differences in physical condition according to BMI and fat mass distribution.

These results can serve as evidence and be used in schools in cities with cold climates to reflect on the importance of good sleep hygiene, the rational use of screens, a healthy diet and an active lifestyle in schoolchildren. Likewise, it should be noted that the results of this study could have a positive effect on society in the form of programs and interventions in schools that focus on reducing unhealthy lifestyles and the risk of chronic non-communicable diseases at an early age.

Future research should compare the evaluation of fat mass through electrical bioimpedance or skinfolds against waist circumference in the sub-classification of obesity to assess the accuracy of a quick, economical, and accessible method. In addition, it opens future lines of research concerning the classification of students according to their adiposity measured by an easily accessible indicator.

## 5. Conclusions

A high percentage of schoolchildren self-reported sleep quality problems and spending a lot of screen time daily. Concerning physical fitness, H/HR schoolchildren have lower cardiorespiratory and musculoskeletal fitness than their H/LR and O/LR peers. Coincidentally, a higher percentage of schoolchildren in the O/HR group were perceived to have poor or very poor physical conditions compared to the other groups.

## Figures and Tables

**Figure 1 ijerph-19-13690-f001:**
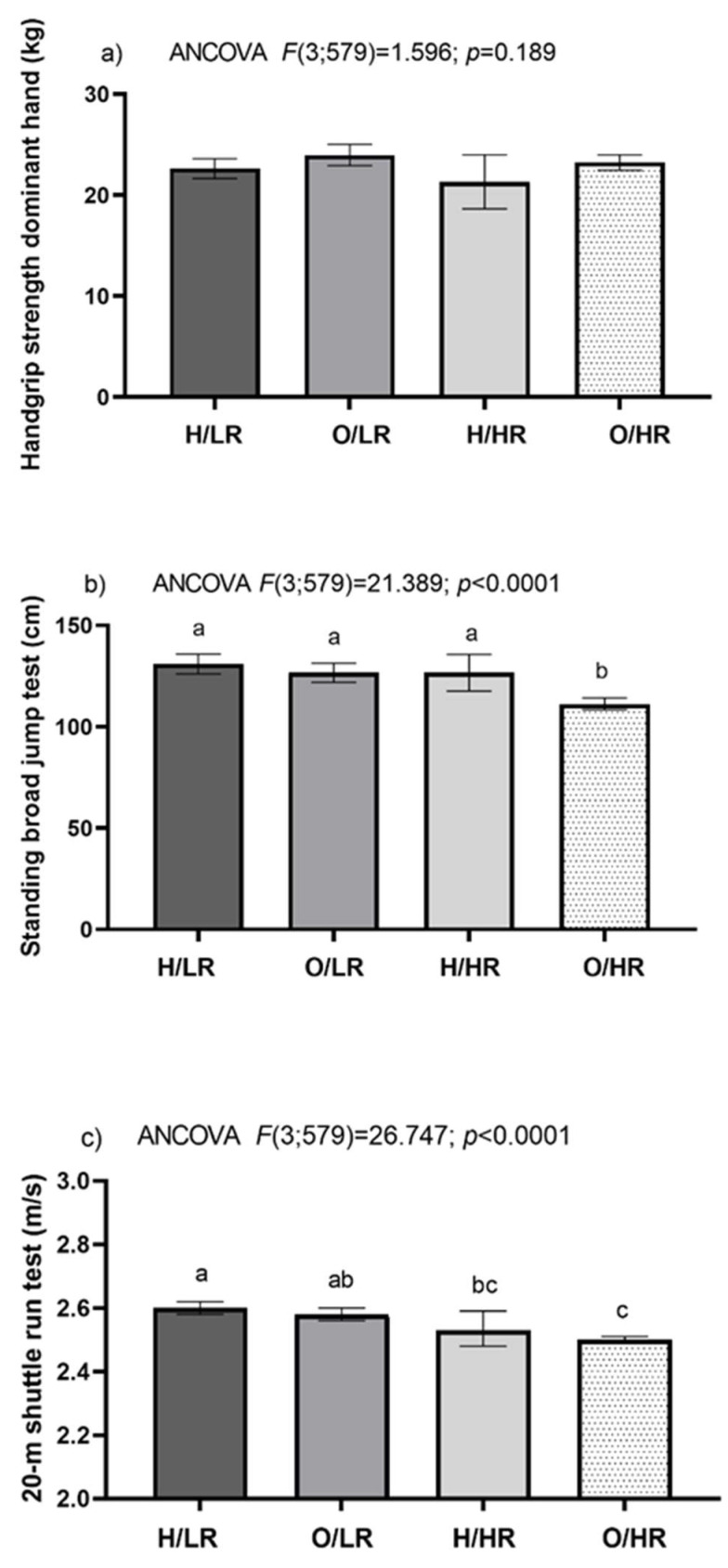
Physical fitness according to sub-classification of obesity. Healthy BMI/low-risk WtHr (H/LR); Overweight and obese/low-risk WtHr (O/LR); Healthy BMI/high-risk WtHr (H/HR); and Overweight and obese/high-risk WtHr (O/HR). In the same row, abc different lowercase letters indicate statistically significant differences between groups (Bonferroni post hoc). *n* = 583.

**Table 1 ijerph-19-13690-t001:** Socio-educational characteristics of the schoolchildren according to obesity sub-classification.

Outcomes	N(%)	H/LR	O/LR	H/HR	O/HR
N(%)		141 (24.2%)	121 (20.8%)	21 (3.6%)	300 (51.4%)
Age (years) M (95% CI)		12.2 (12.0;12,4)	12.2 (11.9;12.5)	12.2 (11.5;12.8)	11.9 (11.8;12.1)
Weight (kg) M (95% CI)		45.0 (43.7;46.3)	57.7 (55.7;59.8)	43.3 (38.3;48.2)	62.7 (61.1;64.3)
Hight (m) M (95% CI)		1.53 (1.51;1.55)	1.56 (1.54;1.57)	1.48 (1.41;1.54)	1.51 (1.50;1.53)
Waist circumference (cm) M (95% CI)		64.3 (63.5;65.2)	68.0 (67.0;69.1)	73.9 (69.8;80.0)	81.3 (80.3;82.3)
**Gender N(%)**					
Male	296 (50.8%)	66 (46.8%)	61 (50.4%)	16 (76.2%)	153 (51.0%)
Female	287 (49.2%)	75 (53.2%)	60 (49.6%)	5 (23.8%)	147 (49.0%)
**Place of residence N(%)**					
Urban	570 (97.8%)	138 (97.9%)	120 (99.2%)	20 (95.2%)	292 (97.3%)
Rural	13 (2.2%)	3 (2.1%)	1 (0.8%)	1 (4.8%)	8 (2.7%)
**Educative center N(%)**					
EC.1	199 (34.1%)	46 (32.6%)	38 (31.4%)	7 (33.3%)	108 (36.0%)
EC.2	175 (30.0%)	34 (24.1%)	44 (36.4%)	9 (42.9%)	88 (29.3%)
EC.3	209 (35.8%)	61 (43.3%)	39 (32.2%)	5 (23.8%)	104 (34.7%)
**Grade N(%)**					
5th grade	162 (27.8%)	33 (23.4%)	27 (22.3%)	6 (28.6%)	96 (32.1%)
6th grade	150 (25.7%)	36 (25.5%)	32 (26.4%)	6 (28.6%)	76 (25.3%)
7th grade	121 (20.8%)	33 (23.4%)	18 (14.9%)	6 (28.6%)	64 (21.3%)
8th grade	150 (25.7%)	39 (27.7%)	44 (36.4%)	3 (14.3%)	64 (21.3%)
**School integration program N(%)**					
Yes	98 (16.8%)	23 (16.3%)	14 (11.6%)	6 (28.6%)	55 (18.3%)
No	485 (83.2%)	118 (83.7%)	107 (88.4%)	15 (71.4%)	245 (81.7%)

Healthy BMI/low-risk WtHr (H/LR); Overweigh or obese/low-risk WtHr (O/LR); Healthy BMI/high-risk WtHr (H/HR); and Overweight or obese/high-risk WtHr (O/HR). *n* = 583.

**Table 2 ijerph-19-13690-t002:** Perception of physical fitness according to sub-classification of obesity.

	N (%)	H/LR N(%)	O/LR N(%)	H/HR N(%)	O/HR (N%)	Chi-Square *p*-Value
**General physical fitness**						
Very poor	21 (3.5%)	2 (1.4%)	4 (3.3%)	0 (0%)	15 (5.0%)	<0.0001
Poor	96 (16.5%)	18 (12.8%)	5 (4.0%)	2 (9.5%)	71 (23.7%)	
Acceptable	204 (35.0%)	41 (29.1%)	46 (38.0%)	6 (28.6%)	111 (36,9%)	
Good	155 (26.6%)	41 (29.1%)	37 (30.7%)	9 (42.9%)	68 (22.7%)	
Very Good	107 (18.4%)	39 (27.7%)	29 (24.0%)	4 (19%)	35 (11.7%)	
**Cardiorespiratory capacity**						
Very poor	28 (4.8%)	1 (0.7%)	2 (1.7%)	0 (0%)	25 (8.3%)	<0.0001
Poor	135 (23.2%)	16 (11.3%)	17 (14%)	2 (9.5%)	100 (33.3%)	
Acceptable	179 (30.7%)	46 (32.6%)	36 (29.8%)	8 (38.1%)	89 (29.7%)	
Good	139 (23.8%)	44 (31.2%)	31 (25.6%)	6 (28.6%)	58 (19.3%)	
Very Good	102 (17.5%)	34 (24.1%)	35 (28.9%)	5 (23.8%)	28 (9.3%)	
**Muscle strength**						
Very poor	22 (3.8%)	4 (2.8%)	4 (3.3%)	0 (0%)	14 (4.7%)	0.094
Poor	115 (19.6%)	22 (15.6%)	23 (19%)	3 (14.3%)	67 (22.3%)	
Acceptable	184 (31.6%)	37 (26.2%)	39 (32.2%)	12 (57.1%)	96 (32%)	
Good	163 (28.0%)	43 (30.5%)	34 (28.1%)	4 (19%)	82 (27.3%)	
Very Good	99 (17.0%)	35 (24.8%)	21 (17.4%)	2 (9.5%)	41 (13.7%)	
**Speed and agility**						
Very poor	26 (4.5%)	1 (0.7%)	7 (5.8%)	0 (0%)	18 (6%)	<0.001
Poor	105 (18.0%)	19 (13.5%)	16 (13.2%)	3 (14.3%)	67 (22.3%)	
Acceptable	202 (34.6%)	44 (31.2%)	36 (29.8%)	6 (28.6%)	116 (38.7%)	
Good	158 (27.1%)	50 (35.5%)	35 (28.9%)	7 (33.3%)	66 (22.0%)	
Very Good	92 (15.8%)	27 (19.1%)	27 (22.3%)	5 (23.8%)	33 (11.0%)	
**Flexibility**						
Very poor	57 (9.8%)	6 (4.3%)	8 (6.6%)	0 (0%)	43 (14.3%)	<0.001
Poor	155 (26.6%)	35 (24.8%)	28 (23.1%)	6 (28.6%)	86 (28.7%)	
Acceptable	160 (27.4%)	42 (29.8%)	29 (24.0%)	5 (23.8%)	84 (28.0%)	
Good	131 (22.5%)	28 (19.9%)	33 (27.3%)	7 (33.3%)	63 (21.0%)	
Very Good	80 (13.7%)	30 (21.3%)	23 (19.0%)	3 (14.3%)	24 (8.0%)	

Healthy BMI/low-risk WtHr (H/LR); Overweight or obese/low-risk WtHr (O/LR); Healthy BMI/high-risk WtHr (H/HR); and Overweight or obese/high-risk WtHr (O/HR). *n* = 583.

**Table 3 ijerph-19-13690-t003:** Schoolchildren’s lifestyles according to sub-classification of obesity.

Variables	N (%)	H/LR N(%)	O/LR N(%)	H/HR N(%)	O/HR N(%)	Chi-Square *p*-Value
**Bedtime routine**						
With routine	504 (86.4%)	122 (86.5%)	108 (89.3%)	17 (81.0%)	257 (85.7%)	0.679
Without routine	79 (13.6%)	19 (13.5%)	13 (10.7%)	4 (19.0%)	43 (14.3%)	
**Sleep-related anxiety**						
Without anxiety	503 (86.3%)	123 (87.2%)	109 (90.1%)	17 (81.0%)	254 (84.7%)	0.432
With anxiety	80 (13.7%)	18 (12.8%)	12 (9.9%)	4 (19.0%)	46 (15.3%)	
**Sleep quality**						
Adequate	449 (77.0%)	100 (70.9%)	99 (81.8%)	17 (81%)	233 (77.7%)	0.188
With problems	134 (23.0%)	41 (29.1%)	22 (18.2%)	4 (19%)	67 (22.3%)	
**Refusal to sleep**						
No	493 (84.6%)	120 (85.1%)	104 (86%)	18 (85.7%)	251 (83.7%)	0.937
Yes	90 (15.4%)	21 (14.9%)	17 (14%)	3 (14.3%)	49 (16.3%)	
**Sleep Self Report (SSR) total score**						
Adequate	486 (83.4%)	116 (82.3%)	107 (88.4%)	18 (85.7%)	245 (81.7%)	0.381
With problems	97 (16.6%)	25 (17.7%)	14 (11.6%)	3 (14.3%)	55 (18.3%)	
**Screen time**						
Low (<2 h/day)	32 (5.6%)	9 (6.6%)	10 (8.4%)	3 (14.3%)	10 (3.4%)	0.054
High (≥2 h/day)	538 (94.4%)	128 (93.4%)	109 (91.6%)	18 (85.7%)	283 (96.6%)	

Healthy BMI/low-risk WtHr (H/LR); Overweight or obese/low-risk WtHr (O/LR); Healthy BMI/high-risk WtHr (H/HR); and Overweight or obese/high-risk WtHr (O/HR). *n* = 583.

**Table 4 ijerph-19-13690-t004:** Screen time, sleep habits, and physical fitness according to sub-classification of obesity.

	H/LR	O/LR	H/HR	O/HR	ANCOVA	
	M (95% CI)	M (I95% CI)	M (95% CI)	M (95% CI)	F	*P* Value
**Screen time**						
TV (hours/day)	1.82 (1.50; 2.13)	1.73 (1.42; 2.03)	2.10 (1.26; 2.19)	2.06 (1.83; 2.29)	1.116	0.342
Video games (hours/day)	3.42 (2.98; 3.89)	3.25 (2.78; 3.72)	3.19 (2.30; 4.09)	3.75 (3.44; 4.07)	1.285	0.280
Internet (hours/day)	4.04 (3.55; 4.52)	3.54 (3.05; 4.02)	2.86 (1.69; 4.02)	3.60 (3.30; 3.91)	1.613	0.185
Total screen time (hours/day)	9.28 (8.38; 10.14)	8.51 (7.63; 9.39)	8.14 (6.26; 10.02)	9.42 (8.86; 9.95)	1.270	0.272
**Sleep habits (Sleep Self-report)**						
Sleeping routines	0.99 (0.77; 1.22)	0.79 (0.54; 1.03)	1.00 (0.39; 1.61)	0.98 (0.83; 1.13)	0.695	0.555
Anxiety related to sleep	2.55 (2.13; 2.96)	2.21 (1.79; 2.62)	2.67 (1.49; 3.84)	2.57 (2.27; 2.86)	0.669	0.571
Sleep quality	4.61 (4.16; 5.06)	4.31 (3.88; 4.74)	4.90 (4.17; 5.64)	4.41 (4.12; 4.71)	0.553	0.646
Refusal to sleep	1.65 (1.38; 1.92)	1.42 (1.10; 1.74)	1.43 (0.70; 2.16)	1.56 (1.36; 1.76)	0.430	0.731
Total Sleep Self Report	9.80 (8.80; 10.80)	8.72 (7.76; 9.68)	10.00 (7.75; 12.25)	9.52 (8.85; 10.19)	0.894	0.444

Healthy BMI/low-risk WtHr (H/LR); Overweight or obese/low-risk WtHr (O/LR); Healthy BMI/high-risk WtHr (H/HR); and Overweight or obese/high-risk WtHr (O/HR). *n* = 583.

## Data Availability

Data are available upon request due to ethical and privacy restrictions.

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
