# Peer review of "Physical Fitness, Screen Time and Sleep Habits According to Obesity Levels in Schoolchildren: Findings from the Health Survey of the Extreme South of Chile"

_ijerph, 2022, doi:10.3390/ijerph192013690_

Round 1
Reviewer 1 Report
Thank you for presenting this paper on an important subject. There is a great need to understand lifestyle in the all-important chronic disease of obesity, especially in children.
The paper needs to be improved from the point of view of language. The structure of the language interferes with the logical flow and the ability of the reader to follow your presentation. There are some specifics in the lines below.
As well as this, the method has not been fully described. It is important to note, especially to understand reporting bias, who delivered the questionnaire and how this was done. Also, the people recording the measurements were not outlined.
The conclusions of the paper needed to be more clearly reported, and some of the conclusions do not fit the results.
Line 32: Statement needs rewriting - obesity is a worry because it is associated with greater burden of disease, and it has been shown to be related to habits etc.
Line 50: I am not sure relevant is the right word here - relevant to what? burden of disease/actions. Maybe better words would be pressing/important/critical.
Line 63: need to define BMI.
Line 73: references?
Line 76: a definition cannot contemplate; it can define or include.
Line 89: concretely not needed, simply state there are few studies.
Line 96: English is awkward, there is no/little evidence about the association of lifestyle with BMI in the south of Chile. (“is surprising” should be omitted)
Line 100: It is important to clearly define the question. Are you comparing physical fitness to screen time to sleep habits, or would it be more correct to say that you are investigating the association of these variables to the different categories of obesity? It would also be helpful to give an understanding to the reader about the geography of Punta Arenas - climate/latitude/etc.
Line 132: (without racing ahead) is redundant, standing start tells us this.
Line 232: This should be rephrased. They did not have good perception; they perceived their general health to be good. You could have good perception but perceive your health to be poor.
Line 236: The statement is that there is a statistically significant difference, however it is not defined in what direction… did the unhealthy weight group perceive heath better or lower than the healthy weight.
Line 281: The main results of this study are that poor exercise tolerance is correlated to obesity and risk. Also, this is in reference to the population that you studied - schoolchildren in the Ponto Arena, you cannot generalise to most schoolchildren.
Line 294: Need to put your results into the context of other studies.
Line 297: this statement relates to other studies, but this study did not set out to study all these variables (e.g., diet) and in fact this study found no statistical significance in poor sleep and screen time.
Line 312: I am not sure you have shown this, as the only real significant difference has been in ability to exercise and not in the lifestyle: sleep, screen time, etc.
Line 338: Sentence does not make sense.
Line 354: This study does not provide national data; it provides local data specific to Punta Arena.
Line 361: This statement is not true. From your data there was no difference statistically between high risk and low risk adiposity groups and the lifestyle variables you mention.
Author Response
Reviewer 1:
Comment 1: Thank you for presenting this paper on an important subject. There is a great need to understand lifestyle in the all-important chronic disease of obesity, especially in children.
The paper needs to be improved from the point of view of language. The structure of the language interferes with the logical flow and the ability of the reader to follow your presentation. There are some specifics in the lines below.
Answer 1: Thank you very much for the comment. In the new version of the manuscript it has been corrected
Comment 2: As well as this, the method has not been fully described. It is important to note, especially to understand reporting bias, who delivered the questionnaire and how this was done. Also, the people recording the measurements were not outlined.
Answer 2: Thank you very much for the comment. In the new version of the manuscript it has been corrected
Comment 3: The conclusions of the paper needed to be more clearly reported, and some of the conclusions do not fit the results.
Answer 3: Thank you very much for the comment. In the new version of the manuscript it has been corrected
Comment 4: Line 32: Statement needs rewriting - obesity is a worry because it is associated with greater burden of disease, and it has been shown to be related to habits etc.
Answer 4: The statement has been modified.
Comment 5: Line 50: I am not sure relevant is the right word here - relevant to what? burden of disease/actions. Maybe better words would be pressing/important/critical.
Answer 5: The word has been modified.
Comment 6: Line 63: need to define BMI.
Answer 6: BMI has been defined.
Llorca-Colomer, F.; Murillo-Llorente, M.T.; Legidos-García, M.E.; Palau-Ferré, A.; Pérez-Bermejo, M. Differences in Classification Standards For the Prevalence of Overweight and Obesity in Children. A Systematic Review and Meta-Analysis. Clin Epidemiol 2022, 14, 1031-1052, doi:10.2147/clep.S375981.
Comment 7: Line 73: references?
Answer 7: Reference has been added.
Albornoz-Guerrero, J.; Carrasco-Marín, F.; Zapata-Lamana, R.; Cigarroa, I.; Reyes-Molina, D.; Barceló, O.; García-Pérez-de-Sevilla, G.; García-Merino, S. Association of Physical Fitness, Screen Time, and Sleep Hygiene According to the Waist-to-Height Ratio in Children and Adolescents from the Extreme South of Chile. Healthcare (Basel) 2022, 10, doi:10.3390/healthcare10040627.
Comment 8: Line 76: a definition cannot contemplate; it can define or include.
Answer 8: The word has been modified.
Comment 9: Line 89: concretely not needed, simply state there are few studies.
Answer 9: The sentence has been modified.
Comment 10: Line 96: English is awkward, there is no/little evidence about the association of lifestyle with BMI in the south of Chile. (“is surprising” should be omitted)
Answer 10: The sentence has been modified.
Comment 11: Line 100: It is important to clearly define the question. Are you comparing physical fitness to screen time to sleep habits, or would it be more correct to say that you are investigating the association of these variables to the different categories of obesity? It would also be helpful to give an understanding to the reader about the geography of Punta Arenas - climate/latitude/etc.
Answer 11: Thank you very much for the comment, the target has been amended. In addition, coordinates and climate of Punta Arenas have been described in the introduction.
Comment 12: Line 132: (without racing ahead) is redundant, standing start tells us this.
Answer 12: Thank you very much for the comment, the statement has been removed.
Comment 13: Line 232: This should be rephrased. They did not have good perception; they perceived their general health to be good. You could have good perception but perceive your health to be poor.
Answer 13: The sentence has been modified.
Comment 14: Line 236: The statement is that there is a statistically significant difference, however it is not defined in what direction… did the unhealthy weight group perceive heath better or lower than the healthy weight.
Answer 14: The sentence has been modified.
Comment 15: Line 281: The main results of this study are that poor exercise tolerance is correlated to obesity and risk. Also, this is in reference to the population that you studied - schoolchildren in the Ponto Arena, you cannot generalise to most schoolchildren.
Answer 15: Punta Arenas has been added to the paragraph. In this way, it is understood that the results were not generalized to all schoolchildren.
Comment 16: Line 294: Need to put your results into the context of other studies.
Answer 16: We believe that our results are presented in the context of other studies (highlighted in yellow). However, some more important aspects are strengthened.
Comment 17: Line 297: this statement relates to other studies, but this study did not set out to study all these variables (e.g., diet) and in fact this study found no statistical significance in poor sleep and screen time.
Answer 17: Thanks for the comment. This sentence has been modified to be more precise with the references used. In addition, reference 48 has been changed (Currently, it is reference 49).
Alqarni, T.A.; Alshamrani, M.A.; Alzahrani, A.S.; AlRefaie, A.M.; Balkhair, O.H.; Alsaegh, S.Z. Prevalence of screen time use and its relationship with obesity, sleep quality, and parental knowledge of related guidelines: A study on children and adolescents attending Primary Healthcare Centers in the Makkah Region. J Family Community Med 2022, 29, 24-33, doi:10.4103/jfcm.jfcm_335_21.
Comment 18: Line 312: I am not sure you have shown this, as the only real significant difference has been in ability to exercise and not in the lifestyle: sleep, screen time, etc.
Answer 18: We agree with the comment. The statement has been rewritten.
Comment 19: Line 338: Sentence does not make sense.
Answer 19: We agree with the comment. The statement is removed.
Comment 20: Line 354: This study does not provide national data; it provides local data specific to Punta Arena.
Answer 20: The sentence has been rewritten.
Comment 21: Line 361: This statement is not true. From your data there was no difference statistically between high risk and low risk adiposity groups and the lifestyle variables you mention.
Answer 21: We do not agree with this comment. We believe that in the section on practical implications we have been clear and precise. It is indicated that the study provides information on screen time, sleep habits and physical condition. This sentence is true. In addition, it is indicated that the study presents the most up-to-date evidence on sleep quality and high screen time in schoolchildren. This statement is also true.
Reviewer 2 Report
This study focuses on a very important public health topic: obesity in schoolchildren. Overall, it is well written, and the goals are clear. The methodological procedures and tools are described in detail, although there were some minor typos that should be addressed. For example, the word “to” before “mean differences” should be deleted in line 214. The presentation of results is acceptable, and the discussion of main findings is objective. However, the conclusions could be improved. In the latter section, listing the results should be avoided and, instead, authors should highlight the scientific, social or even political value of the studie' main findings. Finally, there is a typo error in the title, the word “Heath” lacks the “L”.
Author Response
Reviewer 2
Comment 1: This study focuses on a very important public health topic: obesity in schoolchildren. Overall, it is well written, and the goals are clear.
The methodological procedures and tools are described in detail, although there were some minor typos that should be addressed.
For example, the word “to” before “mean differences” should be deleted in line 214.
Answer 1: Thank you very much for the comment. the word "to" was removed
Comment 2: The presentation of results is acceptable, and the discussion of main findings is objective. However, the conclusions could be improved. In the latter section, listing the results should be avoided and, instead, authors should highlight the scientific, social or even political value of the studie' main findings.
Answer 2: Thank you very much for the comment. Although it should be noted that the practical and theoretical implications have been highlighted in the section above the conclusions (Practical Implications and Future Lines of Investigation). However, this section has been strengthened.
Comment 3: Finally, there is a typo error in the title, the word “Heath” lacks the “L”.
Answer 3: The title has been corrected.
Round 2
Reviewer 1 Report
Thank you for your replies. The paper has been improved to allow for better understanding of your important contribution. There were still a few confusing sentences that need attention:
Line 233: More than 70% of schoolchildren
perceived their condition... or similar would be more concise.
Line 299: However, when in this study there were no significant differences found in screen-time according to subclassification of obesity.
Line 340-344: However, this study is not except for limitations. Thus, in the present study was used a single indicator of adiposity, such as the waist-to-height ratio (WtHr), in the classification of fat distribution. Although waist circumference may be an effective measure in diagnosing central fat [15], it has not been extensively studied for this purpose and could misclassify individuals.
Is confusing because of language structure. I think you mean that this study has limitations, and that you use two measures of adiposity. Needs to be rewritten.
Author Response
Thank you for your replies. The paper has been improved to allow for better understanding of your important contribution. There were still a few confusing sentences that need attention:
Comment 1, Line 233: More than 70% of schoolchildren
perceived their condition... or similar would be more concise.
Answer 1: thank you very much for the comment, the sentence is corrected, now it is more precise
Comment 2, Line 299: However, when in this study there were no significant differences found in screen-time according to subclassification of obesity.
Answer 2: the sentence was amended.
Comment 3, Line 340-344: However, this study is not except for limitations. Thus, in the present study was used a single indicator of adiposity, such as the waist-to-height ratio (WtHr), in the classification of fat distribution. Although waist circumference may be an effective measure in diagnosing central fat [15], it has not been extensively studied for this purpose and could misclassify individuals.
Is confusing because of language structure. I think you mean that this study has limitations, and that you use two measures of adiposity. Needs to be rewritten.
Answer 3: the sentence was amended.